METHODS AND RESOURCES

# A phylogeny-informed characterisation of global tetrapod traits addresses data gaps and biases

**Mario R. Moura**[1,2,3,4]*, Karoline Ceron[5], Jhonny J. M. Guedes[6], Rosana Chen-Zhao[1], Yanina V. Sica[3,4], Julie Hart[4,7], Wendy Dorman[4,8], Julia M. Portmann[3,4], Pamela González-del-Pliego[4,9], Ajay Ranipeta[4], Alessandro Catenazzi[10], Fernanda P. Werneck[11], Luís Felipe Toledo[1], Nathan S. Upham[12], João F. R. Tonini[13], Timothy J. Colston[14], Robert Guralnick[15], Rauri C. K. Bowie[16], R. Alexander Pyron[17], Walter Jetz[3,4]*

1 Departamento de Biologia Animal, Universidade Estadual de Campinas, Campinas, São Paulo, Brazil, 2 Departamento de Biociências, Universidade Federal da Paraíba, Areia, Paraíba, Brazil, 3 Department of Ecology and Evolutionary Biology, Yale University, New Haven, Connecticut, United States of America, 4 Center for Biodiversity and Global Change, Yale University, New Haven, Connecticut, United States of America, 5 Departamento de Biologia, Universidade Federal do Ceará, Fortaleza, Ceará, Brazil, 6 Programa de Pós-Graduação em Ecologia e Evolução, Departamento de Ecologia, Universidade Federal de Goiás, Goiânia, Goiás, Brazil, 7 New York Natural Heritage Program, State University of New York College of Environmental Science and Forestry, Albany, New York, United States of America, 8 Department of Natural Resources and Environmental Sciences (NRES), University of Illinois Urbana-Champaign, Urbana, Illinois, United States of America, 9 Rui Nabeiro Biodiversity Chair, MED Institute, Universidade de Évora, Évora, Portugal, 10 Department of Biological Sciences, Florida International University, Miami, Florida, United States of America, 11 Programa de Coleções Científicas Biológicas, Coordenação de Biodiversidade, Instituto Nacional de Pesquisas da Amazônia, Manaus Amazonas, Brazil, 12 School of Life Sciences, Arizona State University, Tempe, Arizona, United States of America, 13 Department of Biology, University of Richmond, Richmond, Virginia, United States of America, 14 Biology Department, University of Puerto Rico at Mayagüez, Mayagüez, Puerto Rico, 15 Florida Museum of Natural History, University of Florida, Gainesville, Florida, United States of America, 16 Museum of Vertebrate Zoology and Department of Integrative Biology, University of California, Berkeley, California, United States of America, 17 Department of Biological Sciences, The George Washington University, Washington DC, United States of America

* mariormoura@gmail.com (MRM); walter.jetz@yale.edu (WJ)

**Data Availability Statement:** The R-code used for data analysis and figure presentation is available at https://doi.org/10.5281/zenodo.10582070. The

## Abstract

Tetrapods (amphibians, reptiles, birds, and mammals) are model systems for global biodiversity science, but continuing data gaps, limited data standardisation, and ongoing flux in taxonomic nomenclature constrain integrative research on this group and potentially cause biased inference. We combined and harmonised taxonomic, spatial, phylogenetic, and attribute data with phylogeny-based multiple imputation to provide a comprehensive data resource (TetrapodTraits 1.0.0) that includes values, predictions, and sources for body size, activity time, micro- and macrohabitat, ecosystem, threat status, biogeography, insularity, environmental preferences, and human influence, for all 33,281 tetrapod species covered in recent fully sampled phylogenies. We assess gaps and biases across taxa and space, finding that shared data missing in attribute values increased with taxon-level completeness and richness across clades. Prediction of missing attribute values using multiple imputation revealed substantial changes in estimated macroecological patterns. These results highlight biases incurred by nonrandom missingness and strategies to best address them. While there is an obvious need for further data collection and updates, our phylogeny-informed

TetrapodTraits database is available at https://doi.
org/10.5281/zenodo.10530618 and vertlife.org.

**Funding:** We gratefully acknowledge São Paulo
Research Foundation (FAPESP) for grants
supporting MRM (#2021/11840-6 and #2022/
12231-6), LFT (#2016/25358-3), KC (#2020/
12558-0), and RZC (#2022/15247-0);
Coordenação de Aperfeiçoamento de Pessoal de
Nível Superior (CAPES) for the fellowship to JJMG;
Conselho Nacional de Desenvolvimento Científico -
CNPq for research grants in support of FPW
(#311504/2020-5) and LFT (#302834/2020-6); U.
S. National Science Foundation (NSF) for grants
supporting RCKB (DEB-1441652), RAP (DEB-
1441719), and WJ (DEB-1441737 and DEB-
1441719). WJ also acknowledges support from
NASA grants 80NSSC17K0282 and
80NSSC18K0435. This work was partially
supported by E.O. Wilson Biodiversity Foundation
in furtherance of the Half-Earth Project. The
funders had no role in study design, data collection
and analysis, decision to publish, or preparation of
the manuscript.

**Competing interests:** The authors have declared
that no competing interests exist.

**Abbreviations:** DD, data deficient; IQR,
interquartile range; MAR, missing at random;
MCAR, missing completely at random; MNAR,
missing not at random; NRMSE, normalised root
mean square error; PAM, presence–absence
matrix; PMM, predictive mean matching; SES,
standardised effect-size.

database of tetrapod traits can support a more comprehensive representation of tetrapod
species and their attributes in ecology, evolution, and conservation research.

## Introduction

Over the past two decades, biodiversity science has seen a dramatic growth in large-scale
research in ecology, evolution, and conservation biology, enabled by near-global coverage for
study systems such as terrestrial vertebrates, or Tetrapoda (amphibians, reptiles, birds, and
mammals). These efforts usually rely on datasets spanning wide temporal, spatial, and taxo-
nomic scales [1] that ideally are fully harmonised and well curated. Despite terrestrial verte-
brates being a relatively well-known animal group when compared to invertebrates and plants
[2,3], notable gaps persist across various attributes, including fundamental aspects of species
natural history [4,5]. Consequently, trait-based research on biodiversity is often hampered by
spatially and phylogenetically incomplete datasets [6–8].

Understanding causes of missingness is instrumental to advancing biodiversity data cover-
age. For any species attribute, there are observed and unobserved entries, each with a probabil-
ity of being missing [9]. When all entries, whether observed or unobserved, share the same
likelihood of being missing, data are said to be missing completely at random (MCAR). If
missingness affects only observed entries, the data is termed missing at random (MAR). For
example, a depleted digital scale battery makes weighing subsequent specimens impossible in
the field, resulting in MAR data. Data is considered missing not at random (MNAR) when
missingness is tied to unobserved entries, indicating a link to the missing values themselves.
To illustrate, species exclusive to relatively inaccessible habitats, such as the forest canopy, may
be systematically overlooked in field surveys, with their data missingness linked to the occu-
pied microhabitat. These 3 missing mechanisms—MCAR, MAR, and MNAR—can lead to dif-
ferent configurations of the invisible fraction of the trait space [9–11].

While some missing mechanisms (e.g., MCAR, MAR) primarily affect a single attribute, the
underlying cause behind an attribute MNAR can influence multiple variables, resulting in co-
missingness or shared gaps. For example, a species might lack information on multiple eco-
logical aspects due to being known from only a few specimens with no details on where, when,
and how they were found [12]. Similar circumstances apply to rare species or those collected
solely through passive sampling techniques (e.g., pitfall traps), leaving ecological data unob-
served. Indeed, bias is recognised in the availability of trait data for certain taxa, regions, and
traits [4], and missing values for a given variable may be associated with incompleteness in
others. Such congruent or aggregated (as opposed to segregated) patterns in trait missingness
can arise from societal and research preferences for charismatic species [3], easily sampled tax-
onomic groups, or accessible geographical regions [6,13]. Conversely, segregated patterns may
reveal traits and taxa that are challenging to sample or underrepresented.

Despite the continued limitation of sampled data for many attributes in tetrapods, new
methods can help to minimise these gaps and improve our understanding of biodiversity. Past
practices have included the removal of species with missing data or the replacement of missing
values by observed averages, but these strategies may ultimately reduce statistical power and
increase bias [7,10,14]. More recently, growth in large-scale phylogenetic analyses has boosted
the development of methods to increase the accuracy of imputing missing values [15–18].
Among tetrapods, recent large-scale applications of imputation methods include the use of
phylogenetic regression methods and machine-learning techniques to predict missing values

in trait data for amphibians [19], reptiles [19,20], birds [19,21], and mammals [19,21,22], as well as to inform threat statuses for data deficient and non-assessed species [20,23,24].

We leveraged a fast and automated multiple imputation technique with additional data mobilisation to provide a comprehensive database and assessment of key ecological attributes of all extant 33,281 tetrapod species covered in recent fully sampled phylogenies, including 7,238 amphibians [25], 384 chelonians and crocodilians [20], 9,755 squamates and tuatara [26], 9,993 birds [27], and 5,911 mammals [28]. Our assessment covered standardised species-level attributes for taxonomy, body size, activity time, microhabitat, macrohabitat, ecosystem, threat status, biogeography, insularity, environmental preferences, and human influence. Since not all species have genetic data (representing an important source of the remaining uncertainty about their placements in available phylogenies), we also evaluated completeness in genetic sequences [29]. We pinpointed taxa exhibiting pronounced shared missingness in natural history data to inform new strategies for data acquisition and mitigate biases in trait databases. To enhance database consistency, we taxonomically harmonised data sources and filled gaps using a phylogeny-based multiple imputation method [30–32] for which we verified the performance and associated uncertainty.

We use this gap-filled database to assess the geographic, taxonomic, and trait-related biases and evaluate how their model-based closure supports improved information and biological inference. Due to the biodiversity knowledge paradox—high biodiversity in the tropics [33] but better taxonomic sampling in temperate regions [4,5,34]—we expect larger unsampled fractions in the tetrapod trait space for tropical species. Similarly, the high research capacity (i.e., infrastructure and expertise availability) dedicated to birds and mammals relative to amphibians and reptiles [2,3] contributes to the uneven sampling of trait space [4], likely producing larger biases among historically undersampled taxa. Finally, species biology and sampling methodologies are known to affect detection and collection rates in the field [35–37]. For example, detectability is typically lower for small- than large-bodied species, and similarly so for nocturnal relative to diurnal taxa [38–40], whereas sampling methodologies often favour the collection and research of species living on the surface compared to fossorial or arboreal groups [34,41–44]. We thus anticipate convergent missingness across trait space in tetrapods and expect undersampled species to typically be small, nocturnal, and fossorial or arboreal.

## Methods

We curated and assembled available databases for global tetrapod groups and used the latest phylogeny-based methods to create the most comprehensive tetrapod attribute dataset to date. While the TetrapodTraits database also covers a wide range of attributes derived from species range maps (see S1 Table, Supporting information), our focus regarding the imputations primarily centred on natural history traits, specifically: body length, body mass, activity time, and microhabitat. Our procedures can be summarised in five general steps: (i) data acquisition; (ii) taxonomic harmonisation; (iii) outlier verification; (iv) taxonomic imputation; and (v) phylogenetic multiple imputation.

### Data acquisition

We compiled species-level attributes regarding taxonomy, body size, activity time, microhabitat, macrohabitat, ecosystem, threat status, biogeography, insularity, environmental preferences, and associated data sources for each tetrapod species (S1 Table). We gathered information from several global, continental, and regional databases, and complemented the existing data from published (articles, book chapters, and field guides) and grey literature (e.g., technical reports, government documents, monographs, theses). We also incorporated

unpublished data gathered during fieldwork performed by some of us. To minimise the uneven representation of ecological attributes across clades, we initially identified genera and families whose species did not have available data on body size, activity time, or microhabitat. We then used species belonging to these genera and families to carry out additional online searches on academic platforms (Google Scholar and Web of Science) and included complementary attribute data whenever possible. To improve the chances of finding relevant natural history data [45,46], we conducted these searches using natural history terms in English (e.g., activity time, microhabitat, body size, length, mass, weight), Portuguese (e.g., tempo de atividade, micro-habitat, tamanho de corpo, comprimento, massa, peso), and Spanish (e.g., tiempo de actividad, microhabitat, tamaño del cuerpo, longitud, masa, peso) along with the respective species scientific name or unique synonyms (see Taxonomic harmonisation section). When sources were available in other languages, we employed translation tools for inspection (e.g., Google Translate). In our examination of the data sources, we did not use trait values provided solely at the genus level (e.g., mean value per genus).

Briefly, taxonomic data were represented by higher-level taxonomic ranks (Class, Order, Family), scientific name (same spelling as used in recent fully sampled phylogenies), authority name, and year of description. Three broad natural history traits—body size, activity time, and microhabitat—have been compiled and harmonised across different tetrapod groups. Body size data consisted of information on body length (mm) and body mass (g). Activity time encompassed whether the species was diurnal and/or nocturnal. Cathemeral or crepuscular species were considered as both diurnal and nocturnal. Microhabitat included 5 categories of habitat use commonly reported in field guides and related literature: fossorial, terrestrial, aquatic, arboreal, and aerial.

Microhabitat categories are not mutually exclusive, meaning that a species can be present in more than one category to represent intermediate microhabitats, such as semifossorial (which involves both fossorial and terrestrial categories) or semiarboreal (which combines terrestrial and arboreal). Exceptionally for birds, we adapted microhabitat data from the EltonTraits database [47], which describes the estimated relative usage for seven types of foraging stratum. To make our definition of microhabitat similar across tetrapod groups, we reduced these seven categories to four by: summing the relative usage of species foraging below the water surface or on the water surface in the aquatic microhabitat; summing the relative usage of species foraging on the ground and below 2m in understorey as terrestrial; summing the relative usage of species foraging 2m upward in the canopy and just above canopy as arboreal. Species with aerial microhabitat were kept as defined in EltonTraits database [47], and no fossorial bird was reported in the later source. We then made binary the relative usage of aquatic, terrestrial, arboreal, and aerial microhabitat using a threshold of 30% to consider a species as typical of a given microhabitat type. In a departure from previous mammal databases that treated fossorial and terrestrial species collectively as "terrestrial," we have reviewed microhabitat data to consider fossorial life-style separately from terrestrial [22,47].

Macrohabitat data followed the IUCN Habitat Classification scheme v. 3.1 [48]. This scheme describes 17 major habitat categories in which species can occur and an 18th category for species with unknown major habitat (not included here). We also gathered data on species' major ecosystem (terrestrial, freshwater, and marine). For both macrohabitat and ecosystem, we initially used the *rredlist* package [49] to obtain macrohabitat for 31,740 species and ecosystem for 32,442 species. Our macrohabitat variables correspond only to the first level of IUCN Habitat Classification scheme. For an additional 769 species, we extracted macrohabitat data from relevant literature, bringing the coverage to 32,509 species (97.7% of all species considered). Ecosystem data was extracted from the literature for another 228 species, encompassing 32,670 species and accounting for >98.2% of the total number of species.

We used the *rredlist* package [49] to obtain non-DD assessed status for 29,237 tetrapod species based on IUCN red list v. 2023–1 [50]. For 490 species not available via *rredlist*, we used non-DD assessed statuses matching those described in previous IUCN assessments and included in works using the same taxonomy of fully sampled trees [20,24,26,51]. We also used data on recent published assessment on amphibians [52] and chelonians [53] to inform assessed status for additional 137 species. Across all sources consulted, data deficient species totalled 2,936 species. We did not find an assessed status for 508 species. To enhance the usability of TetrapodTraits, we also provide the respective IUCN binomials for 32,098 species based on IUCN 2023–1 [50].

To compute spatially based attributes in TetrapodTraits, we derived expert-based range maps for amphibians [24,48,54], reptiles [20,33,48], mammals [48,54–56], and birds [27,54]. We matched the authoritative expert range maps for each of the tetrapod groups with the corresponding phylogenies and edited species ranges to ensure that they represented the species concept adopted in the corresponding phylogeny. Overall, our verification procedure of the species range maps can be summarised under 10 scenarios: (i) no changes, where species range maps matched directly with binomials in the phylogeny; (ii) synonyms, where species range maps were direct synonyms to binomials in the phylogeny, thus requiring only an updated name; (iii) split, where species range maps needed to be clipped from a parent species, or when parent species needed to have part of their range removed; (iv) lumps, where species range maps needed to be combined with those of other species; (v) new species-1, where no range map was previously available, so we derived ranges based on recent literature; (vi) new species-2, in the absence of any published map, we drew 10 km radius buffer around point occurrence data (including the species type locality); (vii) new species-3, in the absence of point occurrence data, we drew a polygon around nearby geographical features reported in the literature (e.g., boundaries of a municipality or protected area). We used two additional scenarios for extinct species [56] by referencing the natural ranges of either (viii) extant or (ix) extinct species that coexisted with fossil records of the target extinct species. The last scenario refers to (x) domesticated species, which were represented by their natural ranges before domestication. *Homo sapiens* had its range map represented by the overlapping of all range maps. We did not derive range maps for 11 species (1 amphibian, 2 bats, and 8 squamates) because information on their occurrence was either vaguely defined (e.g., continental land mass or very large administrative unit) or completely absent.

Species expert-based maps were used to compute different attributes related to species range and biogeography. We extracted the latitude and longitude centroids of each range map. Range size was measured as the number of $110 \times 110$ km equal-area grid cells intersected by each species, a spatial resolution that minimises the presence of errors related to the use of expert ranges maps [57–59]. We recorded the presence of a species in a grid cell if any part of the species distribution polygon overlapped with the grid cell. We then computed the proportion of the species range overlapped by each biogeographical realm.

To define if a species was insular endemic or not, we used the literature available [60,61]. We further completed insularity data by registering species whose range maps intersected with minor ($<2$ km$^2$) and major islands worldwide. Island vector data was sourced from Natural Earth (www.naturalearthdata.com, [62]) databases, v. 4.1.0 and v. 5.1.1 for minor and major islands, respectively. Species missing range maps were assumed as non-insular based on the collection of type specimens within major continental land masses (e.g., South Asia, South America, West Africa).

Finally, to inform spatially based attributes we initially extracted the median value of environmental [63,64] and human influence variables [65,66] per grid cell. We then calculated their weighted average within each species range, using the species range occupancy per cell as

weights. This approach aimed to reduce the impact of marginally occupied cells on the attributes derived from within-range measurements [67]. All within-range attributes are individually described in the Results and discussion section (see also S1 Table).

## Taxonomic harmonisation

The taxonomy of the TetrapodTraits database follows the respective taxonomies of the recent, fully sampled phylogenies for each of the major tetrapod groups [20,25–28]. The amphibian phylogeny taxonomy [25] follows the 19 February 2014 edition of AmphibiaWeb (http://amphibiaweb.org), with 7,238 species. The phylogeny for chelonians [20] follows the Turtles of the World (8th ed.) checklist [68], with the addition of *Aldabrachelys abrupta* and *Al. grandidieri*, and the synonymisation of *Amyda ornata* with *Am. cartilaginea*, adding to 357 chelonian species. For crocodilians [20], the taxonomy followed [69], complemented by the revalidation of *Crocodylus suchus* [70] and *Mecistops leptorhynchus* [71], and the recognition of three *Osteolaemus* species [72,73], resulting in 27 species. The taxonomy of the squamate and tuatara phylogeny [26] follows the Reptile Database update of March 2015 (http://www.reptile-database.org) with 9,755 species. For brevity, we refer hereafter to species in the latter phylogeny as squamates, although we recognise that the tuatara, *Sphenodon punctatus*, is not a squamate. The taxonomy of the bird phylogeny [27] followed the Handbook of the Birds of the World [74], including 9,993 species. Finally, the taxonomy of the mammal phylogeny [28] follows the IUCN [75] with modifications resulting in a net addition of 398 species, bringing the total to 5,911 species.

To maximise data usage from previous compilation efforts, and to ensure coherence among species names and the multiple data sources, we built lists of synonyms and valid names based on multiple taxonomic databases [48,76,77], and extracted the unique synonyms in each taxonomic database. By unique synonym, we refer to a binomial (scientific name valid or not) applied to only one valid name. We then performed taxonomic reconciliation based on four steps:

1. ***Direct match with data sources***: We directly paired the names of each of the 33,281 species in TetrapodTraits with the potential source of the data. Species-level attributes of closely related species could appear as identical values if the attributes had been extracted from sources in which different species were treated as synonyms. We minimised the inclusion of duplicated values by flagging each taxonomic match between TetrapodTraits and external data sources to ensure that each data entry was made only once.

2. ***Direct match with data source synonyms***: For the species we were unable to directly match with the data source in step 1, we updated the taxonomy using the list of unique synonyms, and then performed a new matching operation which allowed us to extract and flag additional data whenever possible.

3. ***Direct match with TetrapodTraits synonymies***: Some data sources may follow more recent taxonomies than those inherited from the fully sampled phylogenies [20,25–28]. For species without a direct match after step 2, we updated their taxonomy in the TetrapodTraits using the list of unique synonyms and repeated the data extraction and flagging procedures.

4. ***Manual verification***: For species without a direct match after step 3, we manually searched the specialised taxonomic databases (amphibians [76], reptiles [77], birds [78], and mammals [79]) for potential spelling errors and/or additional synonyms not yet included among our synonym lists. Whenever possible, we updated the taxonomy applied to data sources and then repeated the data extraction and flagging procedure. Species without data after the completion of step 4 were classified as missing data.

## Outlier verification

We implemented two approaches to detect potential inconsistencies in continuous attribute data, body length, and body mass, before applying phylogeny-based methods to impute missing values.

1. *Interquartile range criterion*: Species body length and body mass were $\log_{10}$ transformed and then flagged as outliers if their value were outside the interval defined between $[q_{0.25}-1.5 \times IQR]$ and $[q_{0.75} + 1.5 \times IQR]$, where $q_{0.25}$ and $q_{0.75}$ are respectively the first and third quartiles, and IQR is the interquartile range $[q_{0.25} - q_{0.75}]$.

2. *Deviation from allometric relationship*: Although allometric escape is a phenomenon observed in nature, we used interactive scatterplots to flag species with unusual deviations from the expected allometric relationship between body length and mass. We inspected allometric relationships separately for species within each Class, Order, and Suborder.

For species flagged in steps 1 or 2 above, we checked body length and/or body mass for validity and corrected these values where necessary. Data entries that could not be confirmed using a reliable source were purged from the database.

## Taxonomic imputation

The global scope of the present database inevitably includes some gaps that are hard to fill, e.g., natural history data for species known only from the holotype or a few specimens [12,80]. Previous studies have addressed this challenge and reduced data missingness by using values imputed at the level of genus or from close relatives [47]. Although these earlier strategies of "taxonomic imputation" might artificially reduce variability in attribute values, they are useful for filling gaps in highly conserved attributes, and can ultimately help increase the performance of phylogeny-based imputation methods applied in concert with correlated attribute data [81,82].

We used taxonomic imputations for two cases of missing data in microhabitat: chiropterans, who were considered "aerial" (112 species), and dolphins and whales who were considered "aquatic" (5 species). For the remaining tetrapod species, we computed the per-genus proportion of species in each type of activity time (diurnal or nocturnal), microhabitat (fossorial, terrestrial, aquatic, arboreal, aerial), macrohabitat (17 binary variables informing the IUCN Habitat Classification scheme), and ecosystem (terrestrial, freshwater, marine). If a type of activity time, microhabitat, macrohabitat, or ecosystem appeared in at least 70% of species in the genus, we assumed this ecological attribute was also present among species with missing data in the respective genus. Our goal was to reduce missing values in activity time, microhabitat, and macrohabitat for groups with well-known ecologies (observed data available for at least 70% of species) before running phylogeny-based imputation methods. The number of tetrapod species receiving taxonomic imputations totalled 866 for activity time, 1,110 for microhabitat, 772 for macrohabitat, and 611 for ecosystem. We did not use taxonomic imputation for continuous attributes.

## Phylogenetic multiple imputation

To minimise missing values and capture their uncertainty, we applied the *mixgb* method [30], a recently developed approach that combines the tree-based algorithm *XGBoost* [83] with predictive mean matching (PMM) [84], a multiple imputation technique. *XGBoost* captures interactions and nonlinear relations among variables, while PMM, alongside subsampling, addresses variability associated with missing data. PMM assigns imputed values to each

missing entry based on a group of $k$ donors whose predicted values are the most similar among the observed entries. One donor is then randomly selected, and its observed value is used for imputation [84,85]. The process is repeated $m$ times to produce multiple imputations. When PMM uses a single donor without subsampling, imputations are expected to be identical.

The *XGBoost* does not directly include a phylogenetic tree into its computations. To account for phylogenetic information, we used the phylogenetic covariance matrix of each fully sampled tree [20,25–28] to derive a set of phylogenetic filters (eigenvectors). We determined the number of phylogenetic filters to retain using the broken stick rule [86]. The selection of phylogenetic filters was performed separately for each tetrapod group and across the subset of 100 trees.

To assess the reliability of imputed data, we initially filtered a subset of species with complete data within each tetrapod group. Then, we randomly partitioned these subsets into 10 folds for cross-validation. In each iteration, one fold was excluded from the training process and used as testing data in subsequent modelling. Continuous variables (body length and body mass) were $\log_{10}$-transformed to reduce skewness, while binary variables represented types of microhabitat (fossorial, terrestrial, aquatic, arboreal, aerial) and activity time (diurnal and nocturnal). Note that microhabitat and activity time types are non-mutually exclusive. For birds only, we complemented the observed attribute data with 10 morphometric traits ($\log_{10}$-transformed) made recently available through the AVONET database [87].

*XGBoost* places a central emphasis on the tuning of hyperparameters, covering aspects such as learning rates, tree topology, subsampling, weighting, and regularisation [83,88]. Our tuning procedure began with an initial grid search, exploring 1K parameter combinations uniformly draw from specified ranges for five key hyperparameters: learning rate ($\eta$) from 0.01 to 0.3, maximum tree depth from 3 to 12, subsample portion of training data from 0.7 to 1, minimum child weight from 0.5 to 1.5, and number of boosting iterations (*nrounds*) from 30 to 1,000. For continuous traits, our goal was to minimise the normalised root mean square error (NRMSE) in *XGBoost* regression models using the "reg:squarederror" objective, while for binary traits, we sought to reduce misclassification error in models using the "binary:logistic" objective. We refined the hyperparameter selection by reassessing model performance with an additional 1K parameter combinations uniformly drawn from the parameter ranges defined by the top 5% of models. The tuning procedure was performed separately for each combination of response variable and tetrapod group. Other *XGBoost* parameters were kept at their default values.

Following hyperparameter tuning, we trained *mixgb* models under the 10-fold cross-validation approach. Each *mixgb* model considered the predictive mean matching with 10 donors and provided 10 imputations for each missing entry. The selection of donors, crucial for predictive mean matching in *mixgb* models, is based on exploring the multivariate predictor space (i.e., phylogenetic filters and natural history traits) during the tree building process in *XGBoost* models. Our framework yielded a total of 10K imputed values for each missing entry across the subset of 100 phylogenetic trees (= 10 imputed values × 10 validation folds × 100 phylogenies). In each iteration, we assessed the reliability of imputations using four distinct validation metrics:

1. **Pearson correlation**: computed between imputed and observed values for continuous attributes (body length and body mass).

2. **Regression slope**: computed between imputed and observed values, where slope values $>1$ indicate overestimation, and a slope $<1$ indicates the underestimation for continuous attributes (body length and body mass).

3. **Normalised root mean square error (NRMSE):** computed for continuous attributes (body length and body mass), with lower values indicating higher accuracy [89].

4. **Accuracy:** measured the proportion of correctly classified entries, computed for binary categories representing microhabitat (fossorial, terrestrial, aquatic, arboreal, aerial) and activity time (diurnal and nocturnal).

Overall, the number of tetrapod species receiving phylogenetic multiple imputations totalled 8,123 for body length, 6,752 for body mass, 6,756 for activity time, and 445 microhabitat. All computations were carried out in R version 4.2.3 using the *mixgb* v. 0.1.0 [30], *ape* [90], *gmodels* [91], *hydroGOF* [92], and *stats* [93] packages. Raw data, code, and the 10K imputed values per missing entry per species are reported under data availability [94,95].

In our approach, there are four sources of variability when producing multiple imputed values for each missing data entry. Firstly, the PMM with 10 donors per entry increases variability by avoiding a reduced number of donors. Secondly, the replication of the PMM technique 10 times to potentially select multiple values. Thirdly, the 10-fold cross-validation trained 10 different *mixgb* models per target variable. Finally, we incorporated 100 fully sampled phylogenetic trees, enabling species with imputed evolutionary relationships to vary their position in the multivariate predictor space, guided by phylogenetic filters.

Our goal is to illustrate the utility of multiple imputation to uncover directional bias in natural history data. However, we recognise that this section does not constitute a comprehensive investigation into the impacts of proportion of missing data and the degree of shared missingness on model performance. Delving into this aspect is beyond the scope of the present study and is an area for future research. For further details on mechanisms of data missingness, see [96,97].

## Patterns of shared missing data

We assessed co-occurrence patterns in missing data across species using the "checkerboard score" (C-score; [98]), which is less prone to type II errors than other co-occurrence metrics computed under a null model approach [99]. The C-score was based on a binary presence–absence matrix (PAM) of species (columns) and missing attributes (rows). These attributes were represented by five binary variables informing the absence of observed values in body length, body mass, activity time, microhabitat, and threat status. Missing data in threat status were represented by data deficient (DD) or non-assessed species. We computed the C-score using individual PAMs for each genus and family with at least two species, and each pairwise combination of attribute variables, following recommendations by [100].

To verify whether an observed C-score differed from the value expected by chance, we built a null distribution of C-score values using a randomisation algorithm in which attribute completeness (rows sums) remained fixed while the probability of showing missing attribute values was considered equal for all species. Null distributions were built for each individual PAM using 10K iterations with a burn-in of 500. We then computed the standardised effect-size (SES) of the C-score and associated $p$-values and identified the pairwise attribute combinations with an aggregated (SES C-score $<0$ and $p < 0.05$) or segregated (SES C-score $>0$ and $p < 0.05$) co-occurrence pattern of missing data per taxa. Computations were performed using the *EcoSimR* [101] package in R.

## Patterns and biases in imputed traits

If certain parts of the multivariate attribute space are missing, the inclusion of imputed values may fill such "invisible fractions" and change emergent attribute properties per taxon or

geographical assemblage (e.g., average value, geometric mean, proportion or prevalence of a certain category). But changes in attribute property per taxon can also arise from the inclusion of biased imputed values [81], since imputation errors can increase either with the percentage of missing data or when multiple attributes are imputed simultaneously [15,81,82]. To verify how the inclusion of imputed values affected attribute patterns per taxon, we calculated the per-genus and per-family relative change in attribute property after adding imputed values.

Initially, we extracted the median (for continuous traits, body length, and body mass) or the mean (for binary traits, categories of activity time and microhabitat) value per missing attribute across 10K imputations. We then computed attribute properties per-genus, -family, and -assemblage using (i) observed data only; and (ii) the combined dataset of observed and imputed data. For continuous attributes, we used the geometric mean due to its lower sensitivity to outliers and fluctuations in the median position value. For binary attributes, we computed the mean based on two scores derived from microhabitat and activity time categories: verticality (0 = strictly fossorial, 0.25 = fossorial and terrestrial, 0.5 = terrestrial or aquatic, 0.75 = terrestrial and arboreal, and 1 = strictly arboreal or aerial) and nocturnality (0 = strictly diurnal, 0.5 = cathemeral or crepuscular, 1 = strictly nocturnal). Relative change in attribute property was expressed as the ratio of attribute property obtained with the combined dataset (*obs+imp*) to attribute property with the observed data only (*obs*). We subtracted the relative change from 1 to centre it around zero:

$$RelativeChange = 1 - (AttributeProperty_{obs+imp}/AttributeProperty_{obs})$$

Positive values of relative change indicate the proportional increase in attribute property after inclusion of imputed values, whereas negative values show the opposite. We used the R function *tree.merger* of the *RRphylo* package [102] to build a tetrapod super tree by combining the available global phylogenies for amphibians [25], chelonians and crocodilians [20], squamates [26], birds [27], and mammals [28]. The full tetrapod tree topology was only used for visualisation purposes. In the presence of severe data limitations (e.g., very low completeness, highly uneven sampled data, and lack of inter-correlated attributes), the relative changes in attribute properties could arise due to estimation errors instead of representing unsampled fractions of the attribute space. We therefore used Kruskal–Wallis tests to assess whether the relative increase or decrease of attribute properties differed with respect to the number of simultaneously imputed attributes.

## Results and Discussion

TetrapodTraits offers insights into species-level attributes related to taxonomy, body size, activity time, microhabitat, macrohabitat, ecosystem, geography, environmental preferences, and threat status of tetrapod species (S1 Table). Compilation efforts of previous works have significantly benefited TetrapodTraits, with information extracted from datapapers contributing to 66.6% of the natural history data entries. Additional information, either unpublished or compiled from original sources (e.g., articles, books, grey literature, and websites), accounted for 15.4% of data entries, and imputed values represented 18.1% of the natural history information (Fig 1). In total, data acquisition involved the scanning of more than 3,300 references, including previously published databases.

Imputation performance showed high overall effectiveness across all tetrapods (S1–S3 Figs), particularly for body mass (average Pearson correlation coefficient, r = 0.950, range = 0.692–0.973) and body length (r = 0.933, range = 0.585–0.976). Accuracy was also high for activity time (average = 0.880, range = 0.625–0.886) and microhabitat (average = 0.897, range = 0.742–0.891). Our imputation framework demonstrated a remarkably close match in

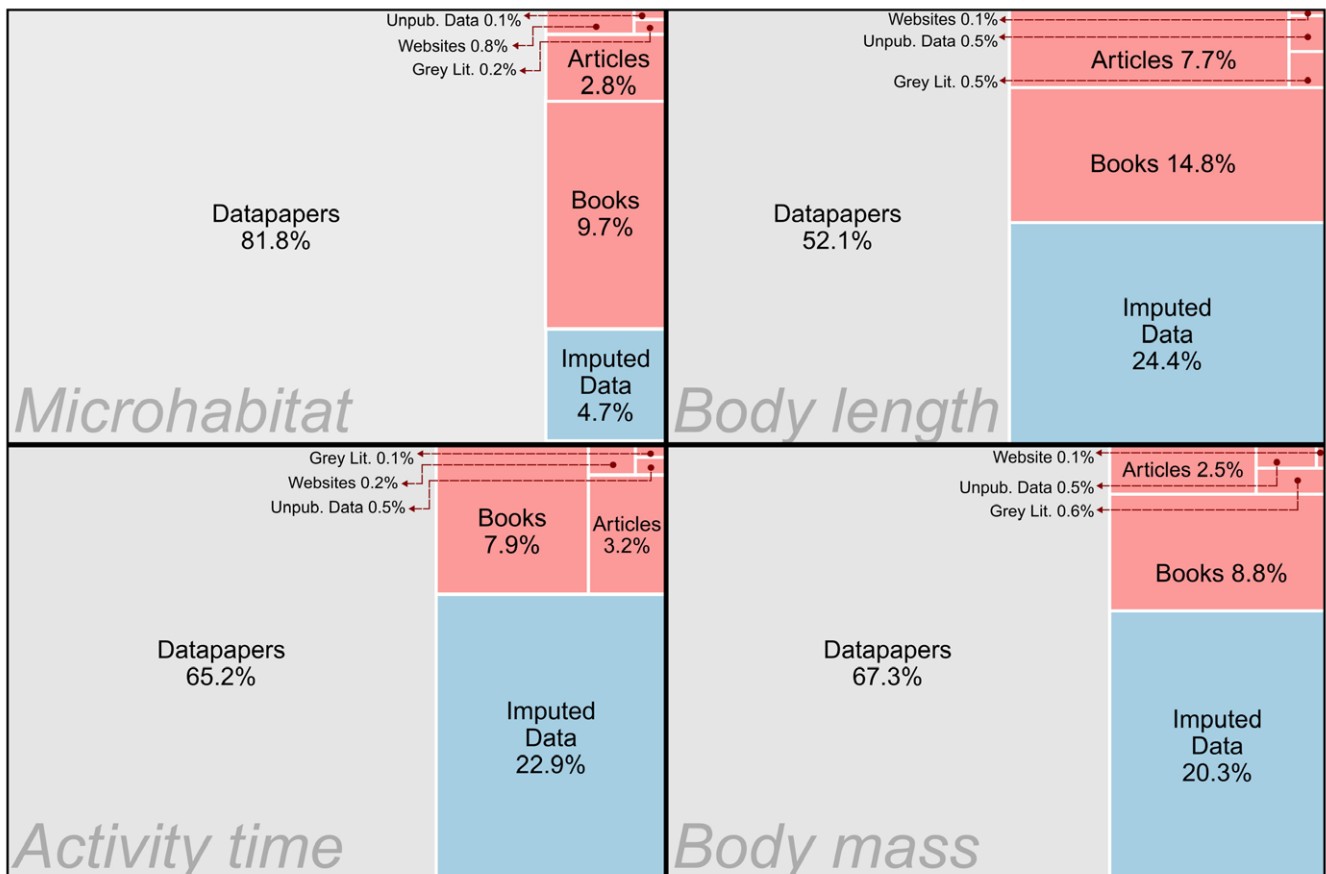

**Fig 1. Proportion of natural history data entries per source type in TetrapodTraits.** To improve readability, the "Grey Lit." boxes aggregate conference proceedings, dissertations, technical reports, preprints, and government documents. "Books" boxes encompass data entries from books, field guides, and book chapters. "Datapapers" boxes represent data entries sourced from datapapers and published articles, featuring raw data compiled from literature either in their appendices or data availability sections. "Articles" boxes represent published articles not classified as datapapers. The data underlying this figure can be found in https://doi.org/10.5281/zenodo.10582069.

frequency distributions between observed and imputed values across the testing datasets, with the multiple imputations effectively capturing the shape, position, and frequency of data entries (S4 Fig). The number of observed, taxonomically imputed, or phylogenetically imputed data entries per tetrapod group are reported in S2 Table.

### Data completeness

Our assessment of data gaps focused on five attributes with considerable missing data: body length, body mass, activity time, microhabitat, and threat status. Across these five attributes, we observed complete species-level data for 43% of tetrapod species ($n = 14{,}321$), 36.9% of genera ($n = 1{,}883$), and 22.4% of families ($n = 116$). No genus or family showed zero completeness for these five attributes combined. The lowest completeness levels were among attributes related to body length (75.6%), activity time (77.1%), body mass (79.7%), and microhabitat (95.3%, S2 Table). While assessed threat status was available for 98.5% of tetrapod species, this number decreased to 89.6% when DD species were treated as missing data (Fig 2).

Across geographic assemblages, missingness of body length showed large variation, with Neotropical and Afrotropical species presenting the lowest completeness (Fig 3A). The low

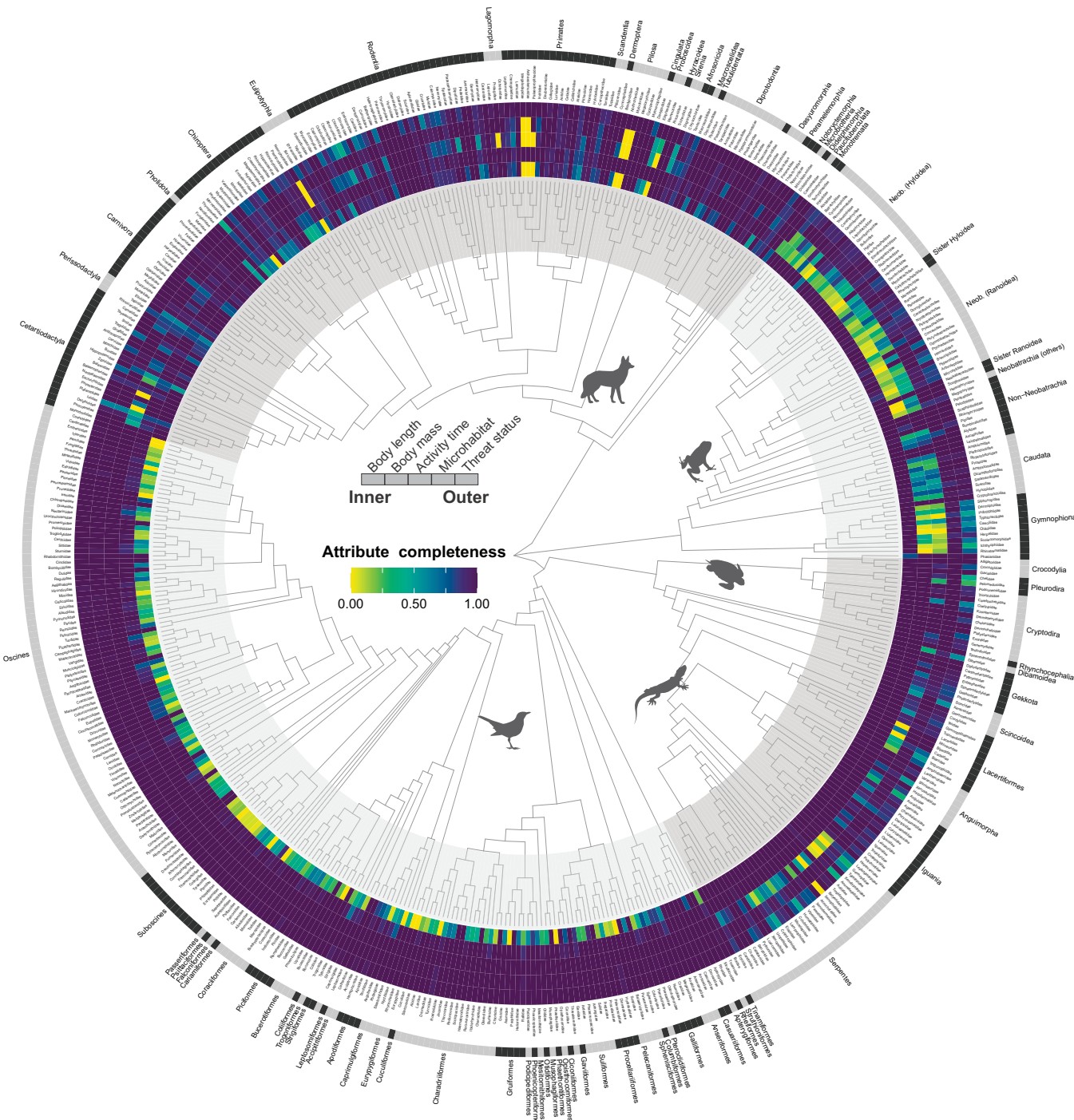

**Fig 2. Attribute data coverage (completeness) of tetrapod species by family.** Colour rings show the proportion of species with observed attributes in each tetrapod family for the 5 attributes with the lowest completeness. Phylogenetic relationships shown are simply the family-level topology of tetrapod families. Darker colours indicate higher completeness. The data underlying this figure can be found in https://doi.org/10.5281/zenodo.10582069.

spatial variability in attribute completeness for body mass, activity time, microhabitat, and assessed threat status (Fig 3B–3E) reveals that species with missing data are mostly narrow-ranged [4], which can limit the influence of these gaps in assemblage-level patterns. Of the 6,233 species missing at least 2 of the 5 attributes (body length, body mass, activity time,

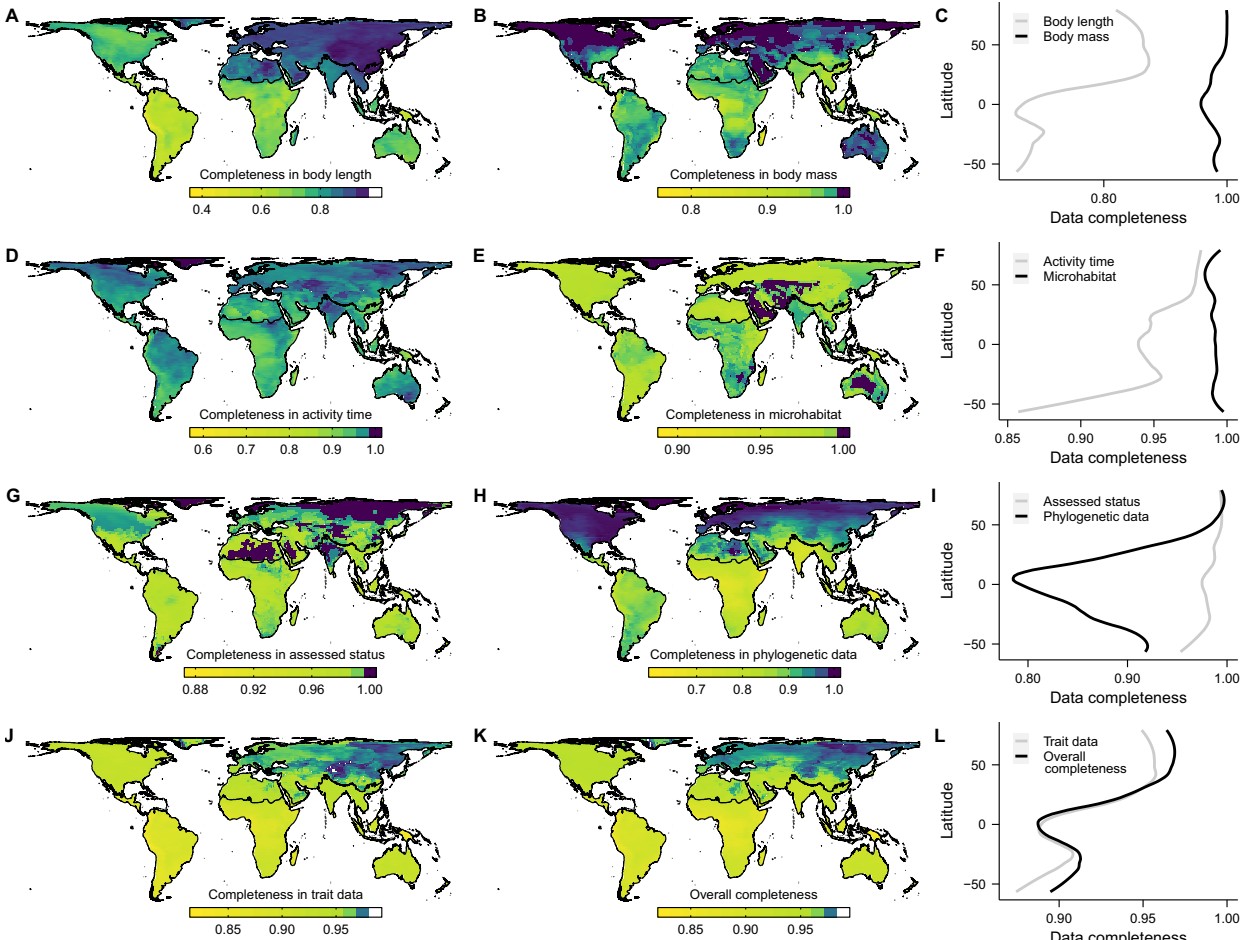

**Fig 3. Average data completeness across tetrapod assemblages.** Proportion of species with observed values for: (A) body length, (B) body mass, (D) activity time, (E) microhabitat, (G) assessed threat status, (H) phylogenetic data, (J) attribute set, i.e., the average pattern for maps depicted in ABDE, and (K) the complete database, i.e., average pattern for maps depicted in ABDEGH. Maps show grid cell assemblages of 110 × 110 km size in an equal-area projection. Latitudinal plots (C, F, I, L) show the average values of these cells across latitudes. Colour ramps followed Jenks' natural breaks classification built separately for each panel. Boundaries of biogeographical realms were adapted from Ecoregions 2017 (https://storage.googleapis.com/teow2016/Ecoregions2017.zip). The data underlying this figure can be found in https://doi.org/10.5281/zenodo.10582069. See S5–S9 Figs for spatial patterns of attribute completeness for each tetrapod group.

microhabitat, and threat status), 77.3% occurred in 10 or fewer 110 × 110 km equal-area grid cells. If we restrict missing data to 3 or more attributes, then 87% of species filtered occur in ≤10 grid cells. The completeness of phylogenetic data in the fully sampled trees of tetrapods [20,25–28] also showed high variation in spatial coverage (Fig 3F), with the Afrotropical and Indomalayan realms emerging with the highest shortfall in phylogenetic data [29]. Overall, our findings help illustrate that geographical patterns in data coverage can be strongly influenced by data for wide-ranged species.

## Patterns of shared missing data

We found nonrandom co-occurrence patterns in data gaps for 15.5% of the pairwise attribute combinations at the genus-level and for 37% at the family-level. Almost all nonrandom co-occurrence patterns showed aggregation of missing values, that is, missing data was shared among multiple attributes for the same species, particularly among squamates and amphibians

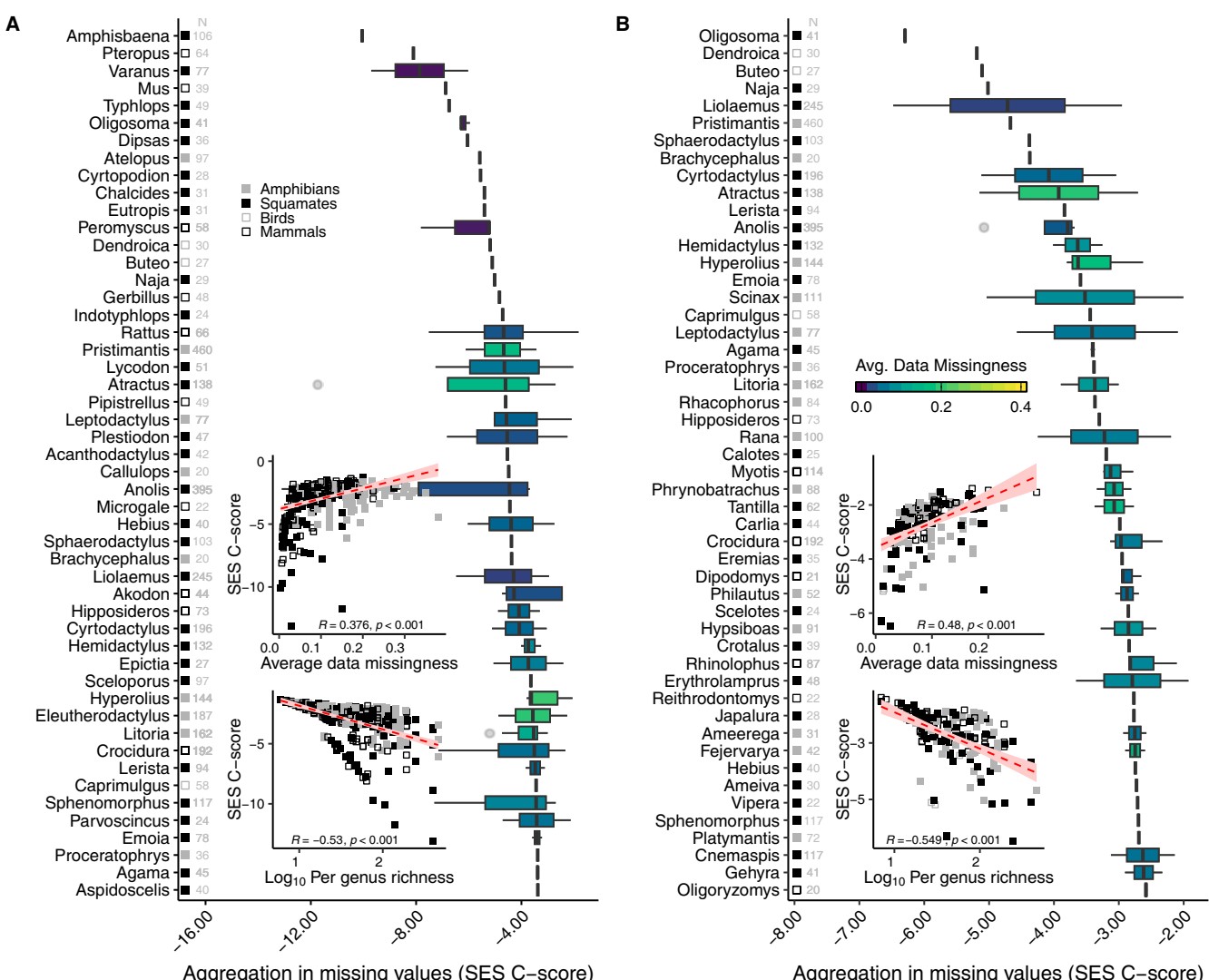

**Fig 4. The top 50 tetrapod genera with the most pronounced patterns of shared data gaps.** For each genus, the aggregation metric equals the median value of the SES of the C-score metric computed across (A) all pairwise attribute combinations among the 5 focal traits and (B) all attribute pairs involving threat status. Grey numbers on the left side of each panel indicate the species richness per genus. The data underlying this figure can be found in https://doi.org/10.5281/zenodo.10582069. See S10 Fig for aggregation patterns computed at the family-level.

(Fig 4A). This supports the perspective that amphibians and squamates are the least researched tetrapod classes [2,3,34]. In our examination of shared gaps between threat status and other attributes, we found higher co-missingness in several tetrapod genera representative of leaf litter-dwelling (*Craugastor*, *Eleutherodactylus*, *Lerista*, and *Pristimantis*), fossorial and semifossorial (*Atractus*, *Crocidura*, and *Tantilla*), and arboreal or scansorial (*Cyrtodactylus*, *Dendroica*, and *Sphaerodactylus*) species. These findings indicate the uneven sampling coverage of attribute space within certain clades.

Aggregated data gaps were more common among species-rich clades with a lower proportion of missing data (Figs 4 and S9). That is, shared gaps were strongest in species-rich and well-sampled taxa, particularly in those holding many species known only from their type locality or holotype [80,103]. About 80% of the genera and 66% of families within the top 50

taxa in shared gaps include "lost taxa," species that have not been reliably observed in >50 years but are not yet declared extinct [12]. While the rediscovery of "lost" species is somewhat common [103], their absence in attribute databases will not be easily filled through field observations [104]. The data gaps resulting from these poorly known species often prevent their inclusion in trait-based research, potentially harming biogeographical research and conservation practice.

## Biases uncovered by imputation-based gap filling

We found that the use of imputations to fill gaps resulted in substantial changes in aggregate attribute properties and the insights they could support. These changes occurred in both directions (increasing and decreasing values) and were strongest in continuous attributes (body length and body mass). After the inclusion of imputed values, the average attribute value per genus decreased in 6.1% of the attribute-genus combinations and increased in 5.3% of these combinations. At the family-level, after the inclusion of imputed attributes, 16.1% of the attribute-family combinations showed an increase in the average values, whereas 19.3% decreased (Fig 5).

Changes in attribute properties at the level of geographic assemblages revealed substantial alterations in average body size in tropical regions (Fig 6A–6F). Such differences are mostly absent in more temperate regions, reflecting the dedicated mobilisation and the resulting large availability of natural history data throughout North America, Europe, and Asia [2,4,105]. After the inclusion of imputed values, there was a notable increase in the average nocturnality of tetrapod assemblages, indicating a significant global bias in knowledge resulting from the absence of observed activity time data (Fig 6H and 6I), particularly for squamates and to a lesser extent, amphibians (S11–S15 Figs). We attribute this to both the low detectability of nocturnal species [39,40] and to the general lack of data for tropical regions where nocturnal species predominate [106]. Changes in average verticality were less pronounced across tetrapod assemblages (Fig 6K and 6L), which is somewhat expected given the relatively high completeness of microhabitat data (S16 Fig).

Although imputation biases can increase in the absence of additional correlated attributes [81], we found no relationship between the number of imputed attributes and changes in attribute averages after imputation-based gap filling (S17 Fig). That is, changes in average attributes were not higher in genera and families that included species with multiple missing attributes. Moreover, changes in average attributes were either mostly unrelated to shared gaps (S18 Fig) or occurred in opposite directions, with low shared gaps leading to greater changes.

Broadly, we found that the magnitude of change in average attributes from imputation-based gap filling was associated with both completeness and richness (S19–S21 Figs). Those taxa and assemblages with great incompleteness and with fewest species saw the most pronounced changes in average attribute values. Lower attribute completeness creates more opportunities for variation in attribute space after imputation-based gap filling, whereas low species richness can enhance the relative importance of imputed values in the sample. Future research replacing imputed values with empirical data will reveal the accuracy of these estimates over time.

## TetrapodTraits—A comprehensive, imputation informed attributes database

The full phylogenetically coherent database we developed, TetrapodTraits, is being made publicly available to support a range of research applications in ecology, evolution, and conservation and to help minimise the impacts of biased data in this model system. The database

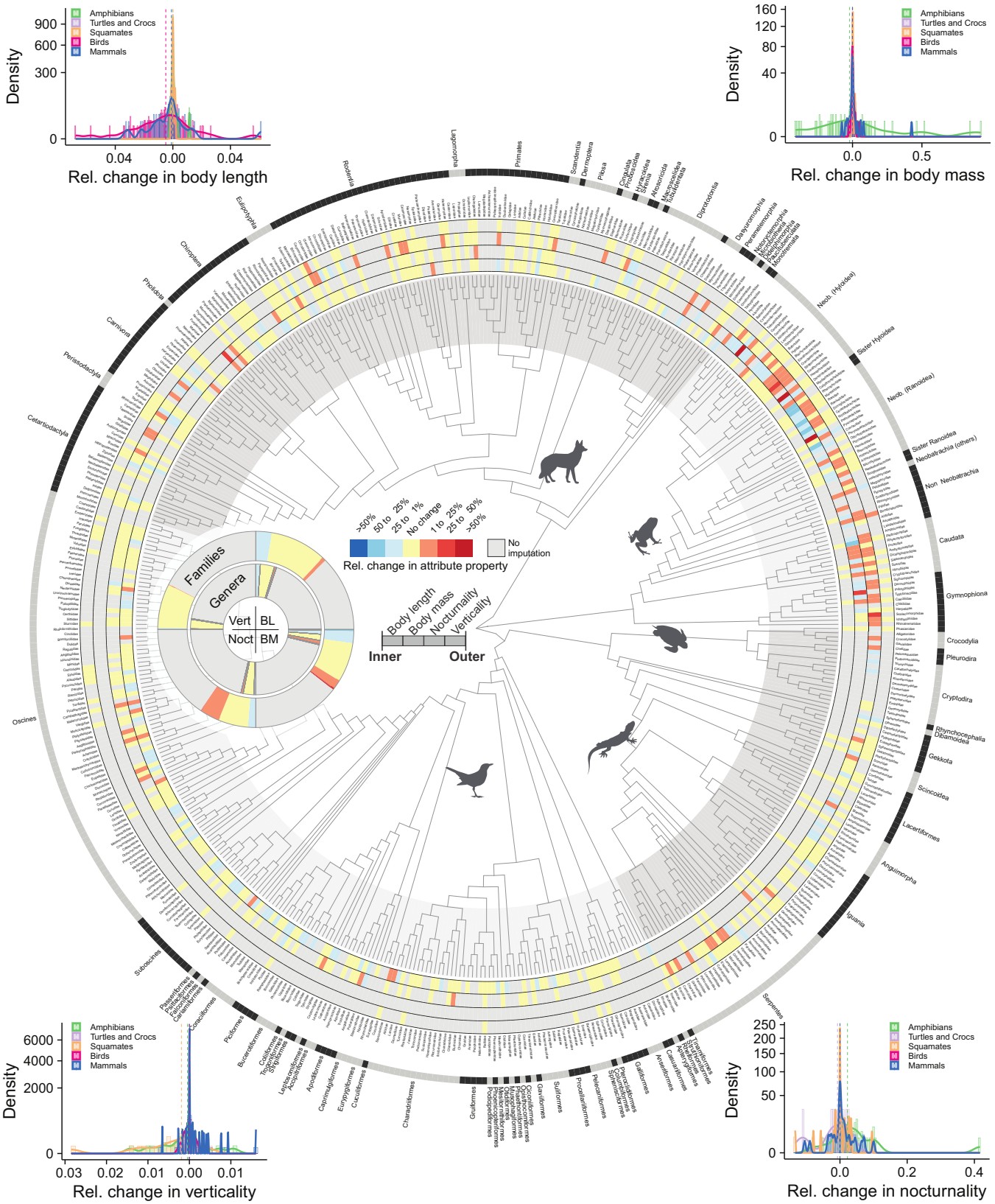

**Fig 5. Changes in family-level attributes values after imputation-based gap-filling.** For continuous attributes (BL, body length; BM, body mass), relative change in attribute property quantifies the proportional change in the geometric mean attribute per genus. For binary attributes (types of activity time and microhabitat), relative change in attribute property measures the change in mean species nocturnality (Noct) and verticality (Vert) scores. The inset donut chart denotes the proportion of genera (inner donut) and families (outer donut) by category of attribute change. The y-axis of histograms is square-rooted to improve readability. The data underlying this figure can be found in https://doi.org/10.5281/zenodo.10582069.

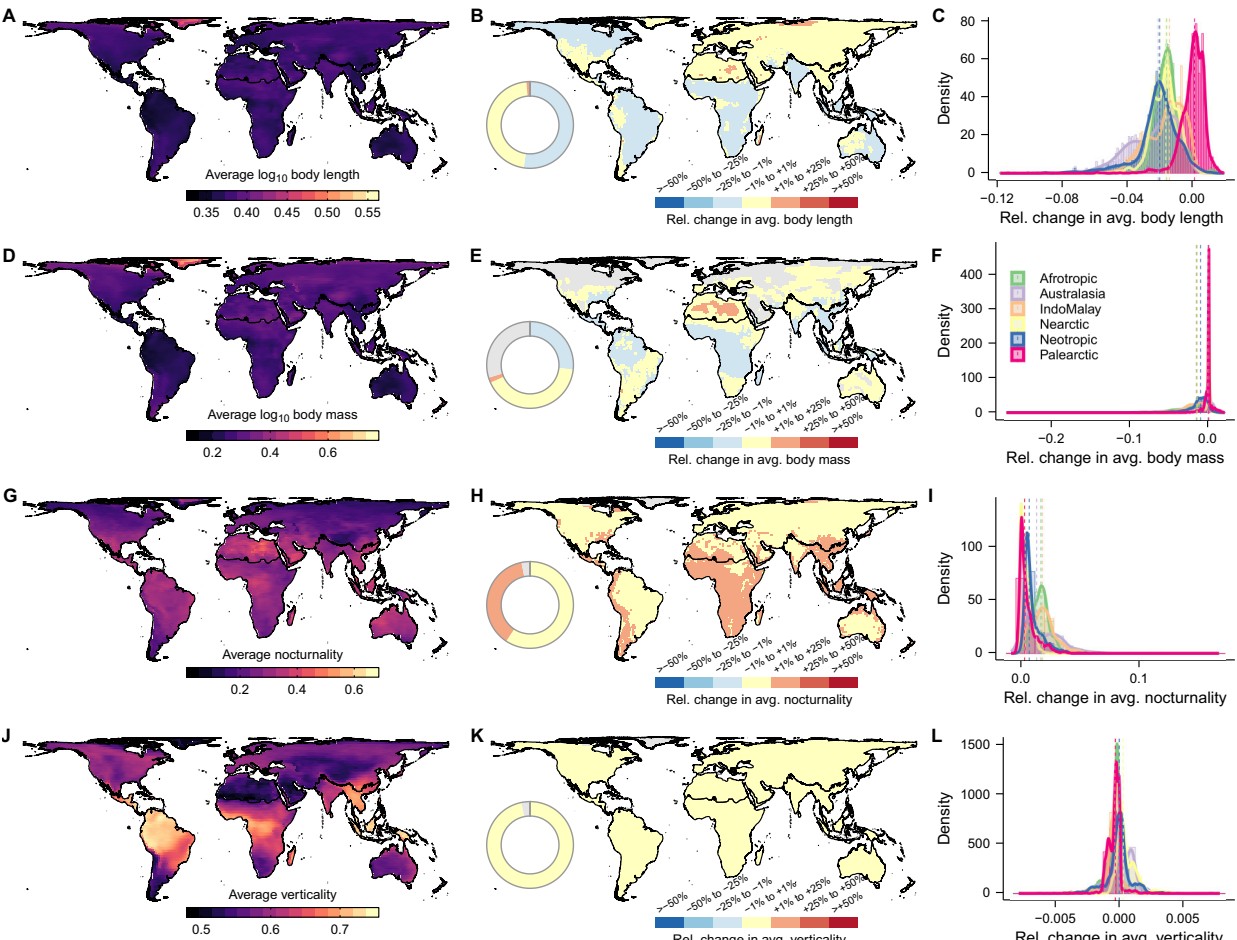

**Fig 6. Changes in average attribute per tetrapod assemblage after imputation-based gap-filling.** For each grid cell, maps show the average species attribute value and respective relative change in attribute value after inclusion of imputed values for (A–C) body length, (D–F) body mass, (G–I) nocturnality, and (J–L) verticality. Body length and body mass were $log_{10}$ transformed before computations. Grey cells indicate assemblages without species with imputed values. Boundaries of biogeographical realms were adapted from Ecoregions 2017 (https://storage.googleapis.com/teow2016/Ecoregions2017.zip). The data underlying this figure can be found in https://doi.org/10.5281/zenodo.10582069. See S11–S15 Figs for patterns of attribute change for each tetrapod group.

includes 24 species-level attributes linked to their respective sources across 33,281 tetrapod species. Entries across this database show at least 98% of completeness after the inclusion of imputed values. Specific fields clearly label data sources and imputations in the TetrapodTraits (S1 Table), while additional tables record the 10K values per missing entry per species.

1. **Taxonomy**–includes 8 attributes that inform scientific names and respective higher-level taxonomic ranks, authority name, and year of species description. Field names: Scientific. Name, Genus, Family, Suborder, Order, Class, Authority, and YearOfDescription.

2. **Phylogenetic tree**–includes 2 attributes that notify which fully sampled phylogeny contains the species, along with whether the species placement was imputed or not in the phylogeny. Field names: TreeTaxon, TreeImputed.

3. **Body size**–includes 7 attributes that inform length, mass, and data sources on species sizes, and details on the imputation of species length or mass. Field names: BodyLength_mm,

LengthMeasure, ImputedLength, SourceBodyLength, BodyMass_g, ImputedMass, SourceBodyMass.

4. **Activity time**–includes 5 attributes that describe period of activity (e.g., diurnal, nocturnal) as dummy (binary) variables, data sources, details on the imputation of species activity time, and a nocturnality score. Field names: Diu, Noc, ImputedActTime, SourceActTime, Nocturnality.

5. **Microhabitat**–includes 8 attributes covering habitat use (e.g., fossorial, terrestrial, aquatic, arboreal, aerial) as dummy (binary) variables, data sources, details on the imputation of microhabitat, and a verticality score. Field names: Fos, Ter, Aqu, Arb, Aer, ImputedHabitat, SourceHabitat, Verticality.

6. **Macrohabitat**–includes 19 attributes that reflect major habitat types according to the IUCN classification, the sum of major habitats, data source, and details on the imputation of macrohabitat. Field names: MajorHabitat_1 to MajorHabitat_10, MajorHabitat_12 to MajorHabitat_17, MajorHabitatSum, ImputedMajorHabitat, SourceMajorHabitat. Major-Habitat_11, representing the marine deep ocean floor (unoccupied by any species in our database), is not included here.

7. **Ecosystem**–includes 6 attributes covering species ecosystem (e.g., terrestrial, freshwater, marine) as dummy (binary) variables, the sum of ecosystem types, data sources, and details on the imputation of ecosystem. Field names: EcoTer, EcoFresh, EcoMar, EcosystemSum, ImputedEcosystem, SourceEcosystem.

8. **Threat status**–includes 3 attributes that inform the assessed threat statuses according to IUCN red list and related literature. Field names: IUCN_Binomial, AssessedStatus, SourceStatus.

9. **RangeSize**–the number of $110 \times 110$ km grid cells covered by the species range map.

10. **Latitude**–coordinate centroid of the species range map.

11. **Longitude**–coordinate centroid of the species range map.

12. **Biogeography**–includes 8 attributes that present the proportion of species range within each biogeographical realm. Field names: Afrotropic, Australasia, IndoMalay, Nearctic, Neotropic, Oceania, Palearctic, Antarctic [107].

13. **Insularity**–includes 2 attributes that notify if a species is insular endemic (binary, 1 = yes, 0 = no), followed by the respective data source. Field names: Insularity, SourceInsularity.

14. **AnnuMeanTemp**–Average within-range annual mean temperature (Celsius degree). Data derived from CHELSA v. 1.2 [63].

15. **AnnuPrecip**–Average within-range annual precipitation (mm). Data derived from CHELSA v. 1.2 [63].

16. **TempSeasonality**–Average within-range temperature seasonality (Standard deviation $\times$ 100). Data derived from CHELSA v. 1.2 [63].

17. **PrecipSeasonality**–Average within-range precipitation seasonality (Coefficient of Variation). Data derived from CHELSA v. 1.2 [63].

18. **Elevation**–Average within-range elevation (metres). Data derived from topographic layers in EarthEnv [64].

19. **ETA50K** –Average within-range estimated time to travel to cities with a population >50K in the year 2015. Data from [66].

20. **HumanDensity**–Average within-range human population density in 2017. Data derived from HYDE v. 3.2 [65].

21. **PropUrbanArea**–Proportion of species range map covered by built-up area, such as towns, cities, etc. at year 2017 [65].

22. **PropCroplandArea**–Proportion of species range map covered by cropland area, identical to FAO's category "Arable land and permanent crops" at year 2017 [65].

23. **PropPastureArea**–Proportion of species range map covered by cropland, defined as Grazing land with an aridity index >0.5, assumed to be more intensively managed (converted in climate models) at year 2017 [65].

24. **PropRangelandArea**–Proportion of species range map covered by rangeland, defined as Grazing land with an aridity index <0.5, assumed to be less or not managed (not converted in climate models) at year 2017 [65].

## Conclusions

The growth in mobilised attribute data over the last several decades has enabled significant progress and increased geographic and taxonomic generality for global studies in comparative biology [108–110], macroevolution [111], macroecology [112], and conservation [113–115]. Despite the richness of available data, research outcomes and their interpretation often remained constrained by nonrandomly missing data [6,8,82], and by the lack of standardisation across data dimensions (trait, phylogenetic, spatial) and data sources [116]. We show a clear latitudinal bias in the sampling of phylogenetic data, with limited availability of genetic samples towards the Equator. In concert with taxonomic updates [117,118], these issues have imposed significant limits on multi-taxon investigations and global biodiversity synthesis [119–121]. We documented remaining global data gaps for key traits and then used imputation to fill these gaps, revealing the biases in our knowledge due to the missingness of attribute data. By using phylogeny-based imputation methods in concert with inter-related ecological attributes, we uncovered the hitherto invisible part of attribute space. Our findings confirmed the predominance of attributes that share missing data and provide a comprehensive assessment of gaps and biases across tetrapod groups.

Attributes predictions for species with missing data brought to light key correlates and allowed us to identify the significant consequences these gaps can have for inference. For example, larger body size has been related to both high species detectability [38,39,42] and discovery probability [67,122–124], larger research effort and public interest [34,44,125,126], increased availability of assessed status [127,128], and better data sampling. Despite the difficulties associated with researching large-sized species—e.g., high sensitivity to disturbances [129,130], lower abundance [131,132], challenges with collection or transport—the presence of multiple data biases towards small-sized species confirm the struggle of biodiversity scientists to accurately characterise the smallest organisms and the lower tail of size distributions in nature.

Biodiversity knowledge shortfalls are also clearly related to activity time and microhabitat. Nocturnal species often show lower detectability [39], and typically are less researched [44] and underassessed with respect to threat status [127], which helps explain why diurnal species are more likely to have activity time data. In addition, the higher detectability of terrestrial

than arboreal or fossorial species via standard sampling methods [41,42] can lead to less biased occurrence data [133] and more research effort [34,44]. Although foraging stratum may constrain certain aspects of our biodiversity knowledge, species verticality itself has a limited influence on the availability of microhabitat data.

Attribute imputations can be limited and biased [81,82]. For tetrapods generally and this study specifically, these concerns were minimised by the already substantial taxon-specific datasets and by the gap-oriented mobilisation of additional data. We had to impute all 4 focal attributes (body length, body mass, activity time, and microhabitat) in only 182 tetrapod species (<0.25%). No family or genus had the 4 focal attributes missing in all species. Two important diagnostics offered strong support for the robustness of our attribute predictions. First, there was no association between the number of imputed attributes and changes in average attributes after imputation-based gap filling (S17 Fig). Second, there was no positive association between shared gaps in attribute data and changes in average attribute values (S18–S21 Figs). We encourage researchers to consider how assumptions and uncertainty of any imputed data might affect downstream use and interpretation. However, for many tetrapod research cases, including species with imputed data is preferable to wholesale exclusion of data-limited species.

We have demonstrated that even for Tetrapoda, a central model system in global biodiversity science, our ecological knowledge on attribute data consistently lacks certain types of traits, taxa, and regions. Through careful imputation, we discovered that these gaps have led to attribute distributions with strong geographical and taxonomic biases. We expect that our approach and likely our general insights regarding biodiversity knowledge shortfalls will transcend taxa and systems. The consequences of biased data can percolate through trait-based metrics and comparative analyses, and may ultimately mislead research on functional diversity [8,19,134], extinction risk [6,21,135], and ecogeography [136–138]. The new phylogeny-based attribute database we have constructed can help minimise the impact of biased data and support new avenues of research in tetrapod conservation, ecology, and evolution.

## Supporting information

**S1 Fig. Performance of phylogenetic multiple imputation across natural history traits of tetrapods.** The y-axis indicates the Pearson correlation coefficient for continuous traits (body length and body mass, $\log_{10}$ transformed) and the proportion of correctly classified entries (accuracy) for binary traits (types of activity time and microhabitat). The data underlying this figure can be found in https://doi.org/10.5281/zenodo.10582069.
(PDF)

**S2 Fig. Performance of phylogenetic multiple imputation for continuous attributes.** Validation metrics were computed between imputed and observed $\log_{10}$ attribute values, and include: NRMSE (normalised root mean square error), Pearson (Pearson correlation coefficient), and Slope (linear regression slope). The data underlying this figure can be found in https://doi.org/10.5281/zenodo.10582069.
(PDF)

**S3 Fig. Performance of phylogenetic multiple imputation in classifying activity time and microhabitat types.** The accuracy, measured as the proportion of correctly classified entries, was computed by comparing imputed and observed binary values. Results are reported separately for different types of activity time and microhabitat. The data underlying this figure can be found in https://doi.org/10.5281/zenodo.10582069.
(PDF)

**S4 Fig. Comparison between observed and predicted values across tetrapods.** (A, B) Frequency distribution of data entry values for continuous variables (body length and body mass, $\log_{10}$ transformed). (C–I) Relative frequency of data entries for binary variables representing types of activity time (diurnal and nocturnal) and microhabitat (fossorial terrestrial, aquatic, arboreal, and aerial). The data underlying this figure can be found in https://doi.org/10.5281/zenodo.10582069.
(PDF)

**S5 Fig. Average data completeness across amphibian assemblages.** Proportion of species with observed values for: (A) body length, (B) body mass, (D) activity time, (E) microhabitat, (G) assessed threat status, (H) phylogenetic data, (J) attribute set, i.e., the average pattern for maps depicted in ABDE, and (K) the complete database, i.e., average pattern for maps depicted in ABDEGH. Maps show grid cell assemblages of $110 \times 110$ km size in an equal-area projection. Latitudinal plots (C, F, I, L) show the average values of these cells across latitudes. Colour ramps followed Jenks' natural breaks classification. Boundaries of biogeographical realms were adapted from Ecoregions 2017 (https://storage.googleapis.com/teow2016/Ecoregions2017.zip). The data underlying this figure can be found in https://doi.org/10.5281/zenodo.10582069.
(PDF)

**S6 Fig. Average data completeness across chelonian and crocodilian assemblages.** Proportion of species with observed values for: (A) body length, (B) body mass, (D) activity time, (E) microhabitat, (G) assessed threat status, (H) phylogenetic data, (J) attribute set, i.e., the average pattern for maps depicted in ABDE, and (K) the complete database, i.e., average pattern for maps depicted in ABDEGH. Maps show grid cell assemblages of $110 \times 110$ km size in an equal-area projection. Latitudinal plots (C, F, I, L) show the average values of these cells across latitudes. Colour ramps followed Jenks' natural breaks classification. Boundaries of biogeographical realms were adapted from Ecoregions 2017 (https://storage.googleapis.com/teow2016/Ecoregions2017.zip). The data underlying this figure can be found in https://doi.org/10.5281/zenodo.10582069.
(PDF)

**S7 Fig. Average data completeness across squamate assemblages.** Proportion of species with observed values for: (A) body length, (B) body mass, (D) activity time, (E) microhabitat, (G) assessed threat status, (H) phylogenetic data, (J) attribute set, i.e., the average pattern for maps depicted in ABDE, and (K) the complete database, i.e., average pattern for maps depicted in ABDEGH. Maps show grid cell assemblages of $110 \times 110$ km size in an equal-area projection. Latitudinal plots (C, F, I, L) show the average values of these cells across latitudes. Colour ramps followed Jenks' natural breaks classification. Boundaries of biogeographical realms were adapted from Ecoregions 2017 (https://storage.googleapis.com/teow2016/Ecoregions2017.zip). The data underlying this figure can be found in https://doi.org/10.5281/zenodo.10582069.
(PDF)

**S8 Fig. Average data completeness across bird assemblages.** Proportion of species with observed values for: (A) body length, (B) body mass, (D) activity time, (E) microhabitat, (G) assessed threat status, (H) phylogenetic data, (J) attribute set, i.e., the average pattern for maps depicted in ABDE, and (K) the complete database, i.e., average pattern for maps depicted in ABDEGH. Maps show grid cell assemblages of $110 \times 110$ km size in an equal-area projection. Latitudinal plots (C, F, I, L) show the average values of these cells across latitudes. Colour ramps followed Jenks' natural breaks classification. Boundaries of biogeographical realms were

adapted from Ecoregions 2017 (https://storage.googleapis.com/teow2016/Ecoregions2017.zip). The data underlying this figure can be found in https://doi.org/10.5281/zenodo.10582069.
(PDF)

**S9 Fig. Average data completeness across mammal assemblages.** Proportion of species with observed values for: (A) body length, (B) body mass, (D) activity time, (E) microhabitat, (G) assessed threat status, (H) phylogenetic data, (J) attribute set, i.e., the average pattern for maps depicted in ABDE, and (K) the complete database, i.e., average pattern for maps depicted in ABDEGH. Maps show grid cell assemblages of $110 \times 110$ km size in an equal-area projection. Latitudinal plots (C, F, I, L) show the average values of these cells across latitudes. Colour ramps followed Jenks' natural breaks classification. Boundaries of biogeographical realms were adapted from Ecoregions 2017 (https://storage.googleapis.com/teow2016/Ecoregions2017.zip). The data underlying this figure can be found in https://doi.org/10.5281/zenodo.10582069.
(PDF)

**S10 Fig. The top 50 tetrapod families with most pronounced patterns of shared data missingness.** For each family, the aggregation metric equals the median value of the standardised effect-size (SES) of the C-score metric computed across (A) all pairwise attribute combinations or (B) attribute pairs mandatorily involving threat status. Grey numbers on the left side of each panel indicate the per-family species richness. The data underlying this figure can be found in https://doi.org/10.5281/zenodo.10582069.
(PDF)

**S11 Fig. Changes in average attributes per amphibian assemblage after imputation-based gap-filling.** For each grid cell, maps show the average species attribute value and respective relative change in attribute value after the inclusion of imputed values for (A–C) body length, (D–F) body mass, (G–I) nocturnality, (J–L) verticality. Body length and body mass were log10 transformed before computations. Grey cells indicate assemblages without species with imputed values. Maps draw at the spatial resolution of $110 \times 110$ km in an equal area projection. Boundaries of biogeographical realms were adapted from Ecoregions 2017 (https://storage.googleapis.com/teow2016/Ecoregions2017.zip). The data underlying this figure can be found in https://doi.org/10.5281/zenodo.10582069.
(PDF)

**S12 Fig. Changes in average attributes per chelonian and crocodilian assemblage after imputation-based gap filling.** For each grid cell, maps show the average species attribute value and respective relative change in attribute value after the inclusion of imputed values for (A–C) body length, (D–F) body mass, (G–I) nocturnality, (J–L) verticality. Body length and body mass were log10 transformed before computations. Grey cells indicate assemblages without species with imputed values. Maps draw at the spatial resolution of $110 \times 110$ km in an equal area projection. Boundaries of biogeographical realms were adapted from Ecoregions 2017 (https://storage.googleapis.com/teow2016/Ecoregions2017.zip). The data underlying this figure can be found in https://doi.org/10.5281/zenodo.10582069.
(PDF)

**S13 Fig. Changes in average attributes per squamate assemblage after imputation-based gap filling.** For each grid cell, maps show the average species attribute value and respective relative change in attribute value after the inclusion of imputed values for (A–C) body length, (D–F) body mass, (G–I) nocturnality, (J–L) verticality. Body length and body mass were log10

transformed before computations. Grey cells indicate assemblages without species with imputed values. Maps draw at the spatial resolution of 110 × 110 km in an equal area projection. Boundaries of biogeographical realms were adapted from Ecoregions 2017 (https://storage.googleapis.com/teow2016/Ecoregions2017.zip). The data underlying this figure can be found in https://doi.org/10.5281/zenodo.10582069.
(PDF)

**S14 Fig. Changes in average attributes per bird assemblage after imputation-based gap-filling.** For each grid cell, maps show the average species attribute value and respective relative change in attribute value after the inclusion of imputed values for (A–C) body length, (D–F) body mass, (G–I) nocturnality, (J–L) verticality. Body length and body mass were log10 transformed before computations. Grey cells indicate assemblages without species with imputed values. Maps draw at the spatial resolution of 110 × 110 km in an equal area projection. Boundaries of biogeographical realms were adapted from Ecoregions 2017 (https://storage.googleapis.com/teow2016/Ecoregions2017.zip). The data underlying this figure can be found in https://doi.org/10.5281/zenodo.10582069.
(PDF)

**S15 Fig. Changes in average attributes per mammal assemblage after imputation-based gap filling.** For each grid cell, maps show the average species attribute value and respective relative change in attribute value after the inclusion of imputed values for (A–C) body length, (D–F) body mass, (G–I) nocturnality, (J–L) verticality. Body length and body mass were log10 transformed before computations. Grey cells indicate assemblages without species with imputed values. Maps draw at the spatial resolution of 110 × 110 km in an equal area projection. Boundaries of biogeographical realms were adapted from Ecoregions 2017 (https://storage.googleapis.com/teow2016/Ecoregions2017.zip). The data underlying this figure can be found in https://doi.org/10.5281/zenodo.10582069.
(PDF)

**S16 Fig. Proportion of samples across levels of changes in average attributes after imputation-based gap filling.** Each bar shows the relative number of genera, families, and geographical assemblages facing changes in their average species attribute value after filling missing attributes with imputed values. The data underlying this figure can be found in https://doi.org/10.5281/zenodo.10582069.
(PDF)

**S17 Fig. Relative change in attribute average across increasing number of imputed attributes.** Relative decrease (A, B) or increase (C, D) in attribute metric showed at the (A, C) genus- and (B, D) family-level. Each box denotes the median (horizontal line) and the 25th and 75th percentiles. Vertical lines represent the 95% confidence intervals, and black dots are outliers. Horizontal lines denote the position of 25%, 50%, 75% of relative decrease (light to dark blue) or increase (light to dark red) in attribute metric. Small capital letters denote the results of the Kruskal–Wallis tests for the difference in medians across relative change in average attribute value. The data underlying this figure can be found in https://doi.org/10.5281/zenodo.10582069.
(PDF)

**S18 Fig. Changes in aggregate attributes after imputation-based gap filling in relation to shared missing data.** Relative changes in average attribute value per taxa (geometric mean for body length and mass, and mean for nocturnality and verticality). Each point concerns a taxonomic (A) genus or (B) family. *R* denotes the Spearman correlation coefficient between the

relative decrease (blue) or increase (red) in the average attribute value. Only taxa with aggregated patterns of missing data were used in these plots (median Standardised Effect Size of C-score $\leq -1.96$). More negative values of SES C-Score indicate a higher degree of shared missing data. The data underlying this figure can be found in https://doi.org/10.5281/zenodo.10582069. (PDF)

**S19 Fig. Changes in aggregate attributes after imputation-based gap filling in relation to genus completeness and richness.** Relative changes in average attribute value per genus (geometric mean for body length and mass, and mean for nocturnality and verticality). Each point concerns a combination between the relative change in average attribute for a taxonomic genus. *R* denotes the Spearman correlation coefficient between the relative decrease (blue) or increase (red) in the average attribute and (A) per genus attribute completeness and (B) genus richness. The data underlying this figure can be found in https://doi.org/10.5281/zenodo.10582069. (PDF)

**S20 Fig. Changes in aggregate attributes after imputation-based gap filling in relation to family completeness and richness.** Relative changes in average attribute value per family (geometric mean for body length and mass, and mean for nocturnality and verticality). Each point concerns a combination between the relative change in average attribute for a taxonomic family. *R* denotes the Spearman correlation coefficient between the relative decrease (blue) or increase (red) in the average attribute and (A) per family attribute completeness and (B) family richness. The data underlying this figure can be found in https://doi.org/10.5281/zenodo.10582069. (PDF)

**S21 Fig. Changes in aggregate attributes after imputation-based gap filling in relation to assemblage completeness and richness.** Relative changes in average attribute value per tetrapod assemblage (geometric mean for body length and mass, and mean for nocturnality and verticality). Each point concerns a combination between the relative change in average attribute for a tetrapod assemblage. *R* denotes the Spearman correlation coefficient between the relative decrease (blue) or increase (red) in the average attribute and (A) per assemblage attribute completeness and (B) assemblage richness. The data underlying this figure can be found in https://doi.org/10.5281/zenodo.10582069. (PDF)

**S1 Table. Variables included in the TetrapodTraits database.** (DOCX)

**S2 Table. Number of species with natural history data available per tetrapod group.** (DOCX)

## Acknowledgments

We deeply thank the many authors of the numerous sources consulted during the preparation of this database. This work was possible only because of their valuable efforts.

## Author Contributions

**Conceptualization:** Mario R. Moura, R. Alexander Pyron, Walter Jetz.

**Data curation:** Mario R. Moura, Karoline Ceron, Jhonny J. M. Guedes, Rosana Chen-Zhao, Yanina V. Sica, Julie Hart, Wendy Dorman, Julia M. Portmann, Pamela González-del-

Pliego, Ajay Ranipeta, Alessandro Catenazzi, Fernanda P. Werneck, Luís Felipe Toledo, Nathan S. Upham, João F. R. Tonini, Timothy J. Colston, R. Alexander Pyron.

**Formal analysis:** Mario R. Moura.

**Funding acquisition:** Mario R. Moura, Rauri C. K. Bowie, R. Alexander Pyron, Walter Jetz.

**Investigation:** Mario R. Moura.

**Methodology:** Mario R. Moura.

**Software:** Mario R. Moura.

**Supervision:** Walter Jetz.

**Visualization:** Mario R. Moura, R. Alexander Pyron, Walter Jetz.

**Writing – original draft:** Mario R. Moura.

**Writing – review & editing:** Mario R. Moura, Karoline Ceron, Jhonny J. M. Guedes, Rosana Chen-Zhao, Yanina V. Sica, Julie Hart, Wendy Dorman, Julia M. Portmann, Pamela González-del-Pliego, Ajay Ranipeta, Alessandro Catenazzi, Fernanda P. Werneck, Luís Felipe Toledo, Nathan S. Upham, João F. R. Tonini, Timothy J. Colston, Robert Guralnick, Rauri C. K. Bowie, R. Alexander Pyron, Walter Jetz.

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
