## [Editor Report · Decision Letter 0]

8 Feb 2024

Dear Dr Moura, 

Thank you for submitting the new version of your manuscript entitled "A phylogeny-informed characterisation of global tetrapod traits addresses data gaps and biases" for consideration as a Methods and Resources article by PLOS Biology.

Your revisions have now been evaluated by the PLOS Biology editorial staff, and I'm writing to let you know that we would like to send your submission out for re-review.

However, before we can send your manuscript back out to the reviewers, we need you to complete your submission by providing the metadata that is required for full assessment. To this end, please login to Editorial Manager where you will find the paper in the 'Submissions Needing Revisions' folder on your homepage. Please click 'Revise Submission' from the Action Links and complete all additional questions in the submission questionnaire.

Once your full submission is complete, your paper will undergo a series of checks in preparation for re-review. After your manuscript has passed the checks it will be sent out for review. To provide the metadata for your submission, please Login to Editorial Manager (https://www.editorialmanager.com/pbiology) within two working days, i.e. by Feb 10 2024 11:59PM.

Kind regards,

Roli Roberts

Roland Roberts, PhD

Senior Editor

PLOS Biology

rroberts@plos.org

---

## [Decision Letter · Decision Letter 1]

21 Mar 2024

Dear Dr Moura,

Thank you for your patience while we considered your revised manuscript "A phylogeny-informed characterisation of global tetrapod traits addresses data gaps and biases" for consideration as a Methods and Resources at PLOS Biology. Your revised study has now been evaluated by the PLOS Biology editors, the Academic Editor and two of the original reviewers.

You will see that reviewer #1 is now very positive, and you seem to have allayed their main concerns; s/he asks you to flag the issues of phylogenetic bias, disagrees with your use of NRMSE and PMC as imputation metrics (saying that they’re redundant), and suggests a different approach instead (which s/he implies will be easy to implement). Reviewer #2, however, is not mollified. S/he finds your responses to be somewhat superficial, and complains that the code availability seems to be partial (we would insist full provision of the code, in line with our code availability policy); also s/he thinks that you did not respond fully to their previous query about the sources for the natural history data (e.g. as illustrated in Fig 1). While these issues are probably addressable, reviewers #2's main concern is that the overall paper isn’t providing anything very useful.

IMPORTANT: To resolve this disparity of views, we invited the reviewers to cross-comment on each other's assessments. Interestingly, each professed to empathise with the other's point of view (!). We discussed the reviews and the cross-comments with the Academic Editor, who felt that on balance your Resource would be a useful contribution, and that we should invite a further revision. However, we will need full provision of the code and data, and the Academic Editor emphasised that Figure should be made a "more visually attractive and readable."

In light of the reviews, which you will find at the end of this email, we are pleased to offer you the opportunity to address the remaining points from the reviewers in a revision that we anticipate should not take you very long. We will then assess your revised manuscript and your response to the reviewers' comments with our Academic Editor aiming to avoid further rounds of peer-review, although might need to consult with the reviewers, depending on the nature of the revisions.

**IMPORTANT - SUBMITTING YOUR REVISION**

*Resubmission Checklist*

*Published Peer Review*

*PLOS Data Policy*

Sincerely,

Roli Roberts

Roland Roberts, PhD

Senior Editor

PLOS Biology

rroberts@plos.org

REVIEWERS' COMMENTS:

Reviewer #1:

I congratulate the authors in their attempts to correctly account for missing data using multiple imputation. This was my main criticism in my previous review, and it looks like the authors have done a great job. Providing 10000 multiply-imputed datasets is a great way of allowing researchers to deal with imputation issues, without having to do the imputations themselves, which means the authors have control of the imputation process, and hopefully have done a better job than most researchers could do (due to lack of experience with missing data). In addition, the authors have identified which cells in the data table have been imputed so in an extreme case, researchers could do their own imputation. This is the best of both worlds. I need to point out that the use of predictive mean matching (PMM) for multiple imputation is not a phylogenetically informed procedure unless the probability of accepting a value is proportional to the phylogenetic distance between the imputed and donor species, resulting in a phylogenetic bias towards values from near phylogenetic relatives. This is probably a minor point but the authors should recognise this. I still disagree with the use of NRMSE and PFC. These measures of imputation quality are unnecessary in a multiple-imputation framework as during the imputation we want to increase variability into our imputations. In a multiple imputation framework, we don't want NRMSE and PFC to be too low because we are not trying to replace our missing values with the "true" value (which remains unknown), merely drawing from the approximate posterior distribution of each missing value. So in a multiple-imputation framework, these statistics are unnecessary because they don't measure anything meaningful about the quality of the imputations. A far better approach, which is implemented in the mice package, is to look at the densities of the imputed values compared to the true values, for each variable. We do want the distributions of the imputations to have a similar shape and location to that of the observed values, as departures from this means that our imputations are not enough like our non-missing data to be values that "could" have occurred. This can be a graphical check, and you wouldn't need to look at all 10000 imputations. A random sample of 10 - 20 would suffice. Although the imputations here were not produced by mice, it is a comparatively simple procedure to form a mice-compatible data set and use the densityplot() function in mice. See chapter 6 in Stef van Buuren's book on mice, which is available online: https://stefvanbuuren.name/fimd/ Otherwise, the paper is a good contribution and will form the basis for many new studies.

Reviewer #2:

This is a re-review of the manuscript titled "A phylogeny-informed characterization of global tetrapod traits addresses data gaps and biases".

Overall, I appreciated the time the authors spent in revising the manuscript. However, at this point, I still would say I am unconvinced that this paper represents any kind of novel approach or advance or has significant ability to spur on future research. I actually found the responses to the comments by the Reviewers to be quite superficial, sometimes not really addressing the meat of the question/comment. For example, all because the data are aligned with phylogenies does not mean it represents any kind of advance - this is relatively easy to do with a couple of lines of code to align data with phylogenies.

Speaking of code, if I interpret correctly, the code used is only available in a small subset? AND the data are not currently available for review? I apologize if I'm misinterpreting, but in theory if a reviewer asks for where the data is and that they couldn't find it (even though it is implied it is available at vertlife.org) and then there isn't really a legitimate response, I find this frustrating as a reviewer. Moreover, I suggest the authors have a read of this: https://journals.plos.org/plosbiology/article?id=10.1371/journal.pbio.3002516. 

I did appreciate Figure 1, but this confirms my original point that not all that much is being added here. Plus, the 'other sources' is still vague! "often compiled from field guides and taxonomic-specific literature for tetrapod species". I honestly don't know how to interpret that, and in the original review, it seems quite specific what the question was. In theory, this should be quite easy to quantify and respond to the original reviewer query about.

And then, there still remains the philosophical question of why simply using off-the-shelf imputation methods warrants publication. I would lean towards it doesn't. I do appreciate, however, that the authors did more work on using more reliable imputation methods and making the data available for all imputations. But, returning to the point above, this is relatively simple and if ALL code and a workspace where this was done is not provided, then I simply can not judge it meaningfully, and therefore can not recommend publication of this manuscript in PLoS Biology.

---

## [Editor Report · Decision Letter 2]

3 May 2024

Dear Dr Moura,

Thank you for the submission of your revised Methods and Resources "A phylogeny-informed characterisation of global tetrapod traits addresses data gaps and biases" for publication in PLOS Biology. On behalf of my colleagues and the Academic Editor, Uma Ramakrishnan, I'm pleased to say that we can in principle accept your manuscript for publication, provided you address any remaining formatting and reporting issues. These will be detailed in an email you should receive within 2-3 business days from our colleagues in the journal operations team; no action is required from you until then. Please note that we will not be able to formally accept your manuscript and schedule it for publication until you have completed any requested changes.

IMPORTANT: I will be asking my colleagues to include the following two editorial requests alongside their own: Many thanks for providing the data and code in Zenodo. Please cite the location of the data clearly in all relevant main and supplementary Figure legends, e.g. “The data and code needed to generate this Figure can be found in https://zenodo.org/records/10582070" - also, please could you change the word "characterisation" in the title to the US spelling, i.e. "characterization"?

PRESS: We frequently collaborate with press offices. If your institution or institutions have a press office, please notify them about your upcoming paper at this point, to enable them to help maximize its impact. If the press office is planning to promote your findings, we would be grateful if they could coordinate with biologypress@plos.org. If you have previously opted in to the early version process, we ask that you notify us immediately of any press plans so that we may opt out on your behalf.

Sincerely, 

Roli Roberts

Senior Editor

PLOS Biology

rroberts@plos.org